# Illness (self) management, clinical and functional recovery as determinants of personal recovery in people with severe mental illnesses: A mediation analysis

Bert-Jan Roosenschoon[1,2]* , Mathijs L. Deen[2,3] , Jaap van Weeghel[4] , Astrid M. Kamperman[1] , Cornelis L. Mulder[1,5]

1 Department of Psychiatry, Medical Centre, Epidemiological and Social Psychiatric Research Institute, Erasmus MC University, Rotterdam, Netherlands, 2 Parnassia Academy, Parnassia Psychiatric Institute, Den Haag, Netherlands, 3 Faculty of Social and Behavioral Sciences, Institute of Psychology, Leiden University, Leiden, Netherlands, 4 Department of TRANZO, Tilburg School of Social and Behavioral Sciences, Tilburg University, Tilburg, Netherlands, 5 Antes Mental Health Care, Parnassia Psychiatric Institute, Rotterdam, Netherlands

☯ These authors contributed equally to this work.
* b.roosenschoon@parnassiagroep.nl

**Data Availability Statement:** Data are stored at the institutional database of the Erasmus Medical Centre in Rotterdam, The Netherlands. The

## Abstract

This study analyzed the association between changes over time in illness self-management skills and personal recovery in patients with schizophrenia and other severe mental illnesses and determined the degree to which this association was mediated by changes in clinical and functional recovery. The rationale for the hypothesized directions of association and mediation originated from a recent randomized controlled trial (RCT) on Illness Management and Recovery (IMR), the relations between these concepts suggested in a conceptual framework of IMR, and from the results of three meta-analyses. Moreover, earlier studies indicated the relevance of examining personal recovery as an outcome for people with severe mental illnesses. Outpatient participants' data were used in this RCT (N = 165). Difference scores were constructed for all concepts by subtracting scores measured at baseline (T1) from scores at follow-up measurement (T3). We used mediation analysis to describe pathways between changes in illness management (assessed using the Illness Management and Recovery scale client version) and changes in personal recovery (assessed using the Mental-Health Recovery Measure), mediated by changes in clinical (assessed using the Brief Symptom Inventory) and functional recovery (assessed using the Social Functioning Scale). We applied the baseline data of all concepts as covariates. As inferential tests to determine the significance of the indirect paths, confidence intervals were constructed using bootstrap techniques. The results showed that the improvement in overall illness management was directly associated with improvements in personal recovery (B = .30), and indirectly through improvements in clinical recovery (indirect effect = .13) and functional recovery (indirect effect = .08). The main conclusion is that self-reported illness management appears to be more strongly and directly associated with personal recovery than indirectly via clinical and functional recovery. This analysis supports the relevance of self-

datasets on which the analyses are based are available on request from the Local Ethics Committee of the Erasmus Medical Centre in Rotterdam, due to ethical restrictions and patient confidentiality requirements. To request the data, please contact: Dr Eline Huisjes, research coordinator department of Psychiatry of Erasmus Medical Center via e.huisjes@erasmusmc.nl.

**Funding:** Most of the study was funded by Parnassia Psychiatric Institute (The Hague/ Rotterdam). The remainder was funded by an unconditional educational grant from Janssen-Cilag B.V. (1-4H1ZDZ67890). The role of funding source was: none. The funders had no role in study design, data collection and analysis, decision to publish, or preparation of the manuscript.

**Competing interests:** The authors have declared that no competing interests exist.

management interventions such as IMR for the personal recovery of people with severe mental illnesses.

## Introduction

Patients with serious and persistent mental illnesses (SMI) such as schizophrenia face major challenges in attaining their personal goals and fully participating in society. This is because of their recurrent symptoms, cognitive limitations, lack of social support, and social impediments such as stigma [1, 2]. Psychopharmaceutical treatment can reduce the severity of symptoms and relapse. However, there is also the need for effective psychosocial interventions to provide support for patients in illness self-management. Illness self-management interventions aim to enable people to recover by equipping them with the skills and self-confidence they require to actively recognize and manage their individual health problems [3]. Several illness self-management training programs have been developed including Illness Management and Recovery (IMR). The aim of IMR is to achieve progress in recovery by combining better illness management with the pursuit of personal goals [4]. IMR is currently used in several countries, such as in the US, Europe, and Asia.

Illness self-management concerns people's capabilities to take care of themselves and regain control of their lives [5]. This includes the ability to perform the tasks needed to manage and live successfully with the physical, social, and emotional consequences of a serious and persistent condition [3]. More specifically for people with SMI, it includes the ability to reduce their susceptibility to the illness, and to effectively cope with their symptoms. For example, it is necessary to have knowledge of mental illness to be able to make informed treatment decisions together with professionals, and to have the ability to reach out for social support [5, 6].

Complementary to this, IMR, a psychosocial intervention supporting illness self-management includes providing psycho-education and teaching skills for informed decision-making on treatment; providing cognitive-behavioral training to support behavioral tailored medication adherence; teaching coping skills to deal with symptoms or stress; developing a relapse prevention plan; and improving social support through social skills training [3, 4].

Recovery is a complex and multi-dimensional concept that has been defined in several ways [7–10]. In a typology used in previous studies [4, 11, 12], which was used in the present study, three types of recovery were identified, which are not mutually exclusive, but complementary aspects of recovery [13]. The first type is clinical recovery, which concerns the degree of psychiatric symptomatology [14–16]. The second type is functional recovery which can be defined as the degree of vocational and social functioning, such as acting according to age-appropriate role expectations, the performance of daily living tasks without supervision, engagement in social interactions [17], and the degree of independence with regard to housing [9, 18]. The third recovery type is personal recovery, a term that emerged from people with a lived experience of mental illness and also emphasizes the personal nature of the recovery process [19, 20]. Personal recovery includes several components summarized in the word CHIME [21]: connectedness; hope and optimism about the future; identity; meaning in life; and empowerment.

Changes in illness management scores may be related to changes in personal recovery. Previously, in a review of illness management research, a close relationship between illness management and personal recovery was suggested because the improved management of symptoms, relapses, and stresses of everyday life is critical to developing hope and reaching personal recovery goals [5]. This review was based on an empirical literature review of

educational strategies for illness self-management [5]. Forty RCTs were included. The main findings were that the five previously identified empirically supported strategies for self-management in illness were identified, including psychoeducation, cognitive behavioral interventions, developing a relapse prevention plan, and social and coping skills training [5].

IMR was created based on this review [5]. IMR is a psychosocial program fostering illness self-management in people living with schizophrenia and other SMIs [4]. In practice IMR is a comprehensive structured training program for people with SMI. In our RCT IMR was provided in a group format in weekly 90-min sessions including up to eight people. It lasted on average 12 months [12]. The IMR intervention comprised eleven modules as well as practitioner guides and handouts for the participants. The eleven modules included Recovery Strategies, Basic Facts about Mental Illness, the Stress-Vulnerability Model, Building Social Support, Using Medication Effectively, Drug and Alcohol Use, Reducing Relapses, Coping with Stress, Coping with Persistent Symptoms, Getting Your Needs Met in the Mental-Health System, and Healthy Lifestyles [22]. Each group of patients who received IMR was guided by two trainers, who used a combination of motivation enhancement strategies, educational strategies, and cognitive-behavioral techniques [4]. During the first module—Recovery Strategies—the participants identified their personal goals during the program. For half of each session, some participants worked toward these goals. During the other half of each session, all participants worked on the module subjects with the assistance of handouts [12]. Our pilot study demonstrated that completing the modules required an average of three to four sessions [23].

The aim of IMR training is to improve illness management skills and thereby improve clinical recovery which improves personal and functional recovery. The working of IMR is suggested in the theoretical conceptual framework for the Illness Management and Recovery program [4, 24]. This framework suggests that better illness management leads to better clinical recovery, and that better clinical recovery leads to enhanced personal and functional recovery (S1 Fig).

Based on this conceptual framework, in a previous study using structural equation modeling, we cross-sectionally analyzed the association of five components of illness management with the three types of recovery [11]. This analysis used the baseline data of participants in our RCT on IMR (N = 187) [11, 12]. The results showed that one illness management component —coping—was associated with clinical, functional, and personal recovery. The direct associations between coping and functional and personal recovery were stronger than the indirect associations via clinical recovery. One conclusion of this study was that clinical recovery appeared not to be a prerequisite for functional and personal recovery [11].

The current study was also conducted in the context of our RCT (N = 187) investigating the effects of IMR training in outpatients with SMI [12, 24]. In this RCT multilevel modeling was used to investigate group differences over an 18-month period, comprising 12 months of treatment and six months of follow-up. In this RCT we found a significant effect of IMR + care as usual versus care as usual alone in self-reported overall illness management, as well as in personal recovery over an 18 month period [12]. Moreover, in this RCT, clinical and functional recovery over time significantly improved about as much in both the IMR training group as in the care as usual group [12].

Based on these results, we wondered whether the found changes in illness self-management might be related to the found changes in personal recovery. If this were the case, we wondered whether this association could be (partially) mediated by changes in clinical and functional recovery. Rationales for applying those two mediators in the present analysis were: First, in the proposed working of IMR, clinical recovery was a mediator between illness management on the one hand and personal and functional recovery on the other [4]. Second, in an earlier review and meta-analysis, including 37 RCTs with 5790 participants, illness management

appeared to enhance clinical, personal, as well as functional recovery [3]. Third, in two other earlier reviews and meta-analyses functional recovery was associated with personal recovery [16, 25]. One of these two reviews based on eight studies (N = 1938), found a small but significant effect size (r = .21) [16]. In the other review, including 46 studies, using a total of eight personal recovery measures, showed small positive correlations with social factors (support, work and housing, and functioning; r = 0.23–0.31) [25]. Therefore, the research question arose as to whether functional recovery, as well as clinical recovery, might be a mediator that could explain the relationship between illness management and personal recovery.

Therefore, on the basis of the working of IMR suggested in the conceptual framework [4], the results of our RCT [12], and the results of three reviews and meta-analyses, in this study, we were interested to explore to what degree changes in time in overall illness self-management might be associated with changes in personal recovery. Furthermore, we wanted to identify whether these associations might be (partly) mediated by changes in clinical or functional recovery. This research question might be important for future studies to consider which interventions are relevant to improve personal recovery. In this study, mediation analysis was performed to investigate the indirect and direct relations.

Earlier studies have indicated the importance of improvements in personal recovery for people with SMI, including self-perceived growth and leading a satisfactory life despite the presence of persistent symptoms [8, 12, 19, 26, 27]. Therefore, examining personal recovery as an outcome of interventions in people with SMI is considered increasingly relevant [16, 25, 28, 29].

This led us to the following research question: What is the association of changes over time in illness self-management skills on changes in personal recovery, and to what degree is this association mediated by changes in clinical and functional recovery?

## Methods

### Study setting and data collection

This analysis involved patients with SMI from two mental health care institutions in the greater Rotterdam area, the Netherlands. Participants were from 14 participating community mental health teams from these institutions. The teams provided rehabilitation-oriented clinical case management. This analysis did not require extra assessments of patients as the data from our RCT on IMR [12, 24] were used. Inclusion criteria for this RCT were as follows: age 18–65 years; diagnosis with an SMI such as schizophrenia or a persistent mood disorder with or without comorbid disorders (i.e., substance abuse and personality disorders); receiving outpatient treatment; and being willing and able to provide written informed consent. Exclusion criteria comprised previous participation in IMR training and insufficient Dutch language skills. In line with IRB regulations, participants for this RCT were recruited through clinician referrals for IMR. The selected clients indicated both a willingness to participate in IMR and to be informed about the study. An assistant researcher explained the study objectives and procedures. If participants agreed to participate in the study, written informed consent was obtained [12]. From Oct 25, 2012, to May 2, 2014, 187 study participants were recruited. In this RCT, 116 participants were randomly assigned (3:2 ratio) to receive IMR + Care as Usual (n = 116) or Care as Usual alone (n = 71) [12]. We calculated that this study size was appropriate for this RCT [12].

From Oct 25, 2012, to Feb 1, 2017, per study participant assessments were performed at baseline (T1), after 12 months (post-treatment, T2), and after 18 months (six months of follow-up, T3). As the assessors were blinded to group assignment, the data collection procedure was single-blinded [12]. Only the first author had access to information that could identify

individual participants during data collection. After data collection, an anonymized data set was used.

For the current mediation analysis, the data on self-rated illness self-management, clinical, functional, and personal recovery were used from 165 (88%) participants in this RCT on IMR of whom both assessments at baseline (T1) and at follow-up (T3) were available, irrespective of treatment condition. The scores on the four domains reflected the status of those participants at those time points. With respect to the underlying research question, this sample size allowed us to detect medium sized mediation effects with 80% power [30]. Of the 165 participants in this mediation analysis, 104 (63%) in the original RCT belonged to the experimental condition (exposed to care as usual + IMR), and 61 (37%) belonged to the control group (exposed to care as usual, but unexposed to IMR).

This mediation analysis was conducted in the context of an RCT on IMR. This study was conducted in accordance with the Declaration of Helsinki. The study protocol of our RCT on IMR was approved by the accredited medical ethics trial committee at Erasmus University Medical Center Rotterdam (27/8/2012, METC nr NL38605.078.12) [12]. Due to an administrative oversight, the prospective trial registration was initially overlooked; however, this omission was noticed halfway through the data collection period and the trial was registered with the Netherlands Trial Register (NL4931, NTR5033) [12]. This manuscript is consistent with the reporting guidelines of AGReMA-SF, which is specifically designed for articles reporting mediation analyses [31] (S1 Checklist), and the PLOS ONE Clinical Studies Checklist (S2 Checklist), including the STROBE checklist (S3 Checklist).

## Measures

### Illness (self-) management

Illness (self-) management (IM) was assessed with the IMR scale client version. The IMR scale client version is a composite measure of various components of illness self-management. The 15 items rated on a scale of 1–5 comprise key IM elements, which are also trained in the IMR program, and include progress towards goals, knowledge regarding mental illness, relapse-prevention planning, involvement with significant others, coping with symptoms, medication adherence, substance abuse, and symptom distress [4, 24, 32–34]. The IMR scales have good validity and moderate reliability [33, 35]. To identify and correct discrepancies, the Dutch translation has been independently back translated into English and compared with the original version. Evidence has been provided for the reliability and validity of this Dutch version [24, 34]. For the IMR scale, three subscales were found: 'Coping with Illness Outcome,' 'Knowledge and Goals,' and 'Effective Medication Use/Reduced Alcohol and Drug Abuse' [35]. The total score of the IMR scale client version was used in this analysis.

### Personal recovery

Personal recovery (PR) was assessed with the Mental-Health Recovery Measure (MHRM) [36]. The MHRM, whose 30 items are self-rated on a scale of 0–4, is a composite personal recovery scale that measures self-empowerment, learning and new potentials, and spirituality. The authorized Dutch translation [37] is a reliable measure in terms of internal consistency; convergent and divergent validity are generally acceptable [24, 37, 38].

### Clinical recovery

Clinical recovery (CR) was assessed with the Brief Symptom Inventory (BSI). The BSI is self-rated and has 53 items rated on a scale of 0–4. The authors report good validity, internal

consistency and test-retest reliability for the nine symptomatology dimensions: Psychoticism, Depression, Somatization, Phobic Anxiety, Obsessive Compulsive, Interpersonal Sensitivity, Anxiety, Hostility, and Paranoid Ideation; and also good test-retest reliability for the three Global Indices: global severity index (GSI), positive symptom total (PST), and positive symptom distress index (PSDI) [24, 39, 40]. These psychometric qualities also relate to the Dutch translation of the BSI [41, 42].

## Functional recovery

Functional recovery (FR) was assessed with the Social Functioning Scale (SF-scale). The SF-scale is a self-administered questionnaire that measures social and role functioning with 76 items with varying response formats on the following seven dimensions: social withdrawal, relationships, social activities, recreational activities, independence (competence), independence (performance), and employment. This scale has been described as reliable, valid, sensitive, and responsive to change. [11, 24, 43, 44].

## Statistical analysis

Following our research question, we hypothesized that the improvement of overall illness management had direct and indirect pathways via the improvement in clinical and functional recovery, to the improvement in personal recovery. Mediation analysis is a useful statistical method to examine inferentially such direct and indirect relations [45, 46]. As recommended, our mediation analysis was based on theory, previous research, and logical argument [45].

In this mediation analysis, pathways were described between the improvement in illness (self-) management (ΔIM) and the improvement in personal recovery (ΔPR), mediated by the improvement in clinical (ΔCR) and functional recovery (ΔFR) (Fig 2). Difference scores were constructed for all concepts by subtracting scores measured at T1 from scores measured at T3 [46]. To avoid regression to the mean, we applied the baseline data of all concepts as covariates [46–48]. This corrected the correlation between the difference score and the baseline measurement.

Limiting our sample to only those who participated in IMR did not appear necessary to determine the relationships between the concepts because the study participants in the control group also underwent changes in their illness self-management skills [12]. A larger and more heterogeneous group provides more statistical power and more stable results. Consistent with using participants from both experimental conditions in our analysis, we conducted the analysis entering the treatment condition as a covariate into the mediation models.

The regression-equation for the ΔCR mediator consisted of an intercept, ΔIM, baseline IM, and baseline CR; for the ΔFR mediator the regression-equation consisted of an intercept, ΔIM, baseline IM, and baseline FR; for the ΔPR the regression-equation consisted of an intercept, ΔIM, ΔCR, ΔFR, baseline IM, baseline CR, baseline FR, and baseline PR.

To facilitate comparability, standardized regression coefficients were used. The direct path is the effect of ΔIM on ΔPR (c') from a regression analysis with the mediators as covariates (via *b1* and *b2*). The indirect paths where a combination of parameter estimates from regression analyses with ΔIM as a predictor and each of the mediators as outcome (via *a1* and *a2*) and the parameter estimates regarding the covariates in the aforementioned regression analysis (Fig 1).

As inferential tests to determine the significance of the indirect paths, confidence intervals were constructed by multiplying the regression weights of ΔIM in the first regression and of the mediators in the second. This was executed on 5000 bootstrap samples of our dataset, applying bias correction. For this purpose, the PROCESS macro [46] was used in SPSS 27. In

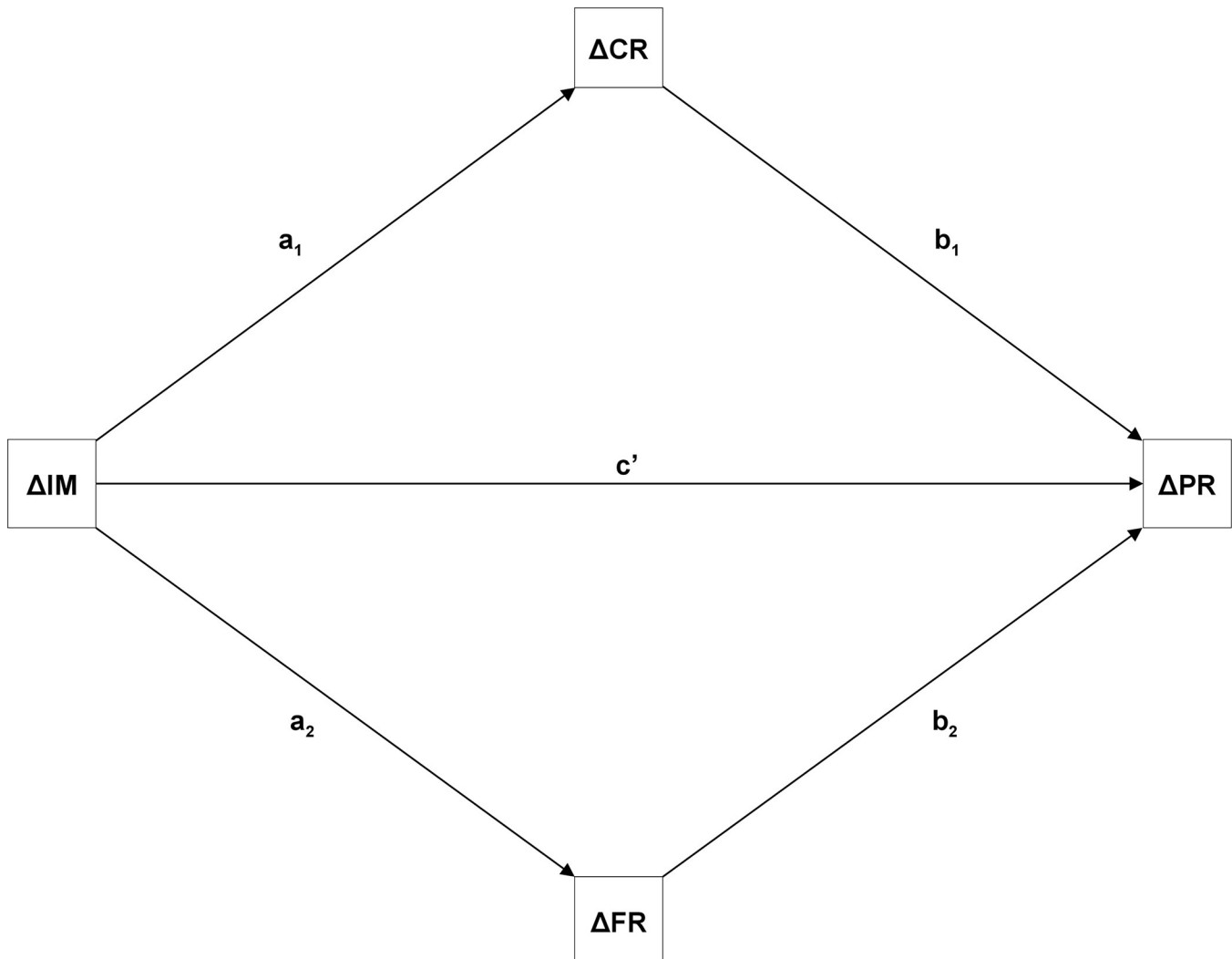

**Fig 1. Parallel multiple mediated path model of the impact of improvement in illness (self-) management (ΔIM) on improvement in personal recovery (ΔPR) mediated by improvement in clinical recovery (ΔCR) and improvement in functional recovery (ΔFR).** *Note*: c', direct impact of ΔIM on ΔPR with mediators partialled out; a1 and a2, impact of ΔIM on mediators ΔCR and ΔFR; b1 and b2, impacts of mediators ΔCR and ΔFR on ΔPR.

this type of mediation analysis, using the PROCESS macro, there can be only one predictor and only one outcome, but multiple mediators can be specified [45, 46].

## Results

### Participants' characteristics

The participants were mostly men who lived alone, had a secondary level of education, were born in the Netherlands, and had a psychotic disorder. The modal treatment length was over five years: they had been admitted at least three times, hospitalized for less than one year, and were receiving income from unemployment, disability, or sick leave (see Table 1).

### Mediation analysis

From a multiple mediation analysis, it appeared that improvements in the overall illness self-management skills of people with SMI strongly directly affected improvements in overall

**Table 1. Participant characteristics.**

| | | N | % |
|---|---|---|---|
| Total | | 165 | 100% |
| Sex | Male | 86 | 52% |
| | Female | 79 | 48% |
| Living situation | | | |
| alone | | 96 | 58% |
| with partner/family | | 44 | 27% |
| in institution[1] | | 25 | 15% |
| Education level | | | |
| Primary | | 61 | 37% |
| Secondary | | 66 | 40% |
| Higher | | 38 | 23% |
| Native country | | | |
| Dutch | | 119 | 72% |
| western immigrant | | 15 | 9% |
| non-western immigrant | | 31 | 19% |
| Source of income | | | |
| Employment | | 10 | 6% |
| benefits for unemployment, invalidity/sickness benefit | | 113 | 69% |
| social security benefit | | 35 | 21% |
| no income | | 5 | 3% |
| Missing | | 2 | 1% |
| Diagnosis[2] | | | |
| psychotic disorders | | 97 | 59% |
| mood disorder | | 53 | 32% |
| personality disorder | | 51 | 31% |
| Length of treatment | | | |
| $\leq$ 5 years | | 37 | 22% |
| > 5 years | | 127 | 78% |
| Missing | | 1 | 1% |
| Number of admissions | | | |
| None | | 40 | 24% |
| 1–2 | | 61 | 37% |
| $\geq$ 3 | | 64 | 39% |
| Length of hospitalization | | | |
| not hospitalized | | 40 | 24% |
| $\leq$ 1 year | | 82 | 50% |
| > 1 year | | 43 | 26% |
| Experimental condition | | | |
| IMR + Care as Usual | | 104 | 63% |
| Care as Usual alone | | 61 | 37% |
| | | M | SD |
| Age (years) | | 44.66 | 10.35 |

[1] sheltered living or in hospital

[2] one person can have more than one diagnosis

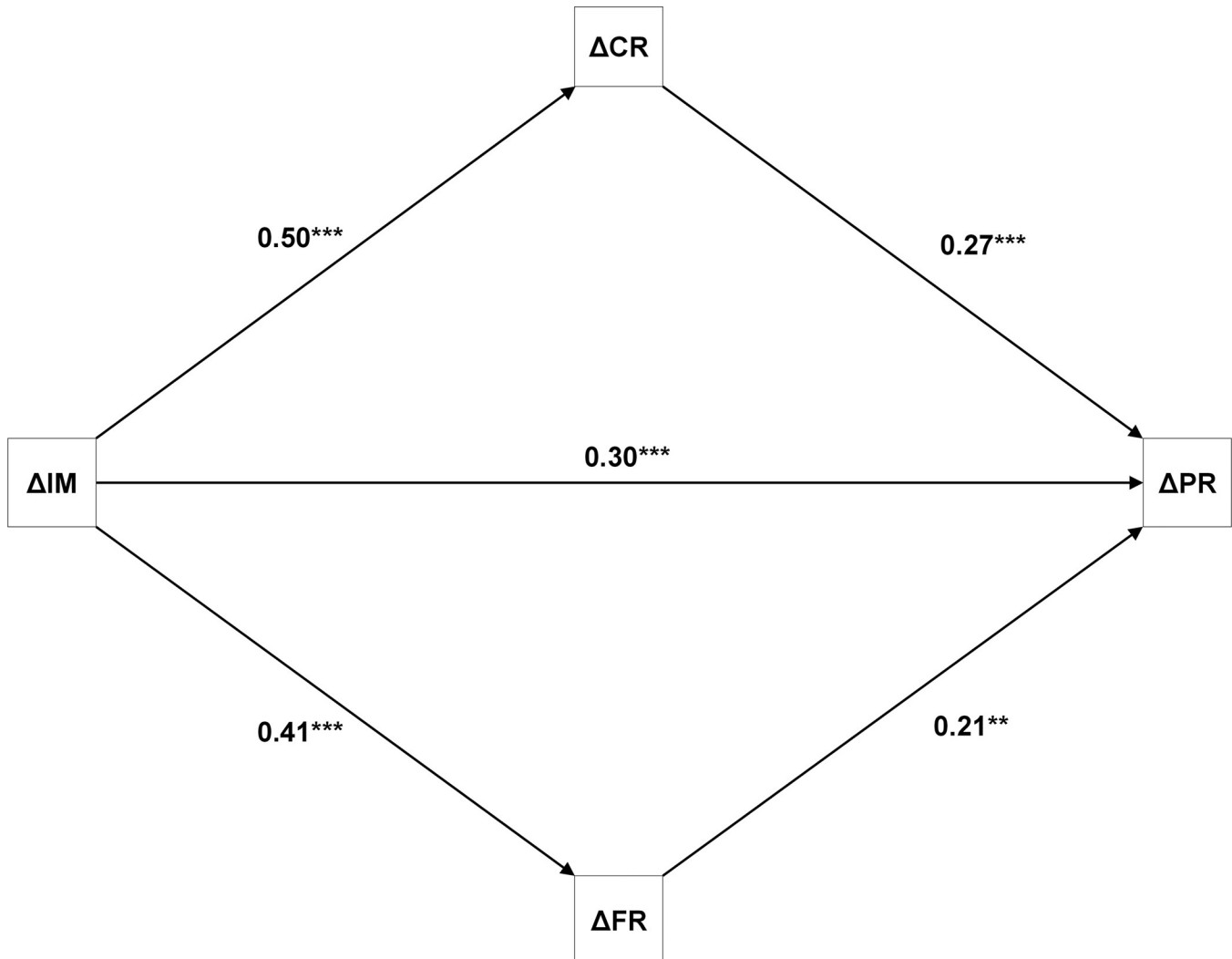

**Fig 2. Path model with standardized regression coefficients of the effect of the improvement of Illness Management (ΔIM) on the improvement of Personal Recovery (ΔPR), and parallel multiple mediated effects of ΔIM on ΔPR via the improvement of Clinical Recovery (ΔCR) and the improvement of Functional Recovery (ΔFR).** *Note*: ***: $p < .001$, **: $p < .01$.

personal recovery independent of its impact on improvements in clinical and functional recovery (c' = 0.30).

It was also shown that improvements in overall illness self-management indirectly affected improvements in overall personal recovery through the effect on improvements in clinical and functional recovery. However, this indirect impact appeared to be less pronounced.

As shown in Fig 2 and Table 2, clients who improved more in overall illness management improved more in clinical and functional recovery than clients who improved less in overall illness management, and clients who improved more in clinical or functional recovery also improved more in personal recovery. Both confidence intervals, of the indirect pathways via improvement of clinical recovery (path $a_1b_1$ = 0.50 x 0.27 = 0.13) and of the pathways via improvement of functional recovery ($a_2b_2$ = 0.41 x 0.21 = 0.08), were above zero, implying statistical significance.

These results suggest that illness self-management appears to be a stronger direct determinant of personal recovery than indirectly via clinical and functional recovery.

**Table 2. Results of multiple mediation analysis of Illness Management (IMRS-client version).**

**on Personal Recovery (MHRM) with bootstrap indirect results**

| Predictor | Path | Estimate | Bootstrap SE | p | Bootstrap 95% CI | |
|---|---|---|---|---|---|---|
| | | | | | Lower bound | Upper bound |
| ΔCR | | | | | | |
| Intercept | | 0.09 | 0.11 | 0.42 | -0.12 | 0.30 |
| ΔIM | a1 | 0.50 | 0.07 | <0.001 | 0.35 | 0.64 |
| IM pre | | 0.34 | 0.08 | <0.001 | 0.18 | 0.51 |
| CR pre | | 0.48 | 0.08 | <0.001 | 0.32 | 0.63 |
| Condition | | -0.14 | 0.14 | 0.32 | -0.41 | 0.13 |
| ΔFR | | | | | | |
| Intercept | | -0.04 | 0.11 | 0.75 | -0.25 | 0.18 |
| ΔIM | a2 | 0.41 | 0.08 | <0.001 | 0.26 | 0.56 |
| IM pre | | 0.45 | 0.09 | <0.001 | 0.28 | 0.62 |
| FR pre | | -0.50 | 0.08 | <0.001 | -0.66 | -0.35 |
| Condition | | 0.03 | 0.14 | 0.81 | -0.24 | 0.31 |
| ΔPR | | | | | | |
| Intercept | | -0.07 | 0.09 | 0.48 | -0.24 | 0.11 |
| ΔIM | c' | 0.30 | 0.07 | <0.001 | 0.16 | 0.44 |
| ΔCR | b1 | 0.27 | 0.07 | <0.001 | 0.14 | 0.41 |
| ΔFR | b2 | 0.21 | 0.07 | <0.01 | 0.08 | 0.34 |
| IM pre | | 0.18 | 0.09 | <0.05 | 0.00 | 0.35 |
| CR pre | | -0.13 | 0.08 | 0.10 | -0.29 | 0.03 |
| FR pre | | 0.17 | 0.08 | <0.05 | 0.19 | 0.32 |
| PR pre | | -0.63 | 0.08 | <0.001 | -0.79 | -0.47 |
| Condition | | 0.13 | 0.11 | 0.24 | -0.09 | 0.36 |
| Indirect effects | | | | | | |
| Total | | 0.22 | 0.05 | | 0.12 | 0.32 |
| ΔCR | a1 x b1 | 0.13 | 0.04 | | 0.06 | 0.22 |
| ΔFR | a2 x b2 | 0.08 | 0.03 | | 0.03 | 0.15 |

*Note*. All outcomes and predictors were standardized; a1, a2, b1, b2 and c' correspond with Fig 1. a x b, bootstrap results for indirect effects; lower and upper limits of bias corrected confidence intervals for test of mediation with 5,000 bootstrap samples.

IM = Illness Management, measured with the IMR scale client version; PR = Personal Recovery, measured with the Mental-Health Recovery Measure; CR = Clinical Recovery, measured with the Brief Symptom Inventory; FR = Functional Recovery, measured with the Social Functioning Scale.

Δ = difference score.

All scores after correction for treatment condition as covariate, and after correction with baseline data of all variables as covariates, indicated with pre.

To enhance the interpretability of the study findings, descriptive statistics pertaining to the study measures at T1 and T3 of the mediation analysis are provided in Table 3.

## Discussion

The research question of this study was: What is the association of changes over time in the illness self-management skills of people with SMI on personal recovery, and to what degree is this association mediated by changes in clinical and functional recovery? Congruent with the formulated hypothesis, the results showed that improvements in self-reported overall illness self-management capabilities were associated with improvements in overall personal recovery in two ways: (1) directly, and (2) indirectly via improvements in clinical and functional recovery. The results suggest that illness (self-) management is an important direct determinant of

**Table 3. Mean (SD) scores on four concepts on T1 and T3 of the mediation analysis (N = 165).**

|  | Mean (SD) | |
| --- | --- | --- |
|  | **T1** | **T3** |
| Illness Self-management (IMRS Client version) | 3.32 (0.49) | 3.49 (0.52) |
| Clinical Recovery (BSI) | 1.25 (0.83) | 1.16 (0.83) |
| Functional Recovery (SFS) | 105.49 (8.64) | 107.17 (9.05) |
| Personal Recovery (MHRM) | 70.55 (20.25) | 72.51 (19.56) |

IMRS = Illness Management and Recovery scale. BSI = Brief Symptom Inventory.

SFS = Social Functioning Scale. MHRM = Mental Health Recovery Measure.

personal recovery, while the indirect effects of illness management mediated by clinical recovery and functional recovery, are less profound. These results also suggest that improvements in clinical and functional recovery may not be prerequisites for improvements in personal recovery.

Our results on the association between illness management and personal recovery are supported by previous findings. An earlier systematic review and meta-analysis showed that the self-management of people with SMI had a significant effect compared with control on subjective recovery measures such as hope and empowerment, as well as on self-rated recovery and self-efficacy [3]. Hope and empowerment are both components of the CHIME conceptual framework of personal recovery [21]. Additionally, our results are supported by a review and meta-analysis of person-oriented recovery interventions which emphasize psychoeducation, developing self-management skills, and fostering self-direction. This meta-analysis involved seven RCTs (N = 1,739), including one on IMR, and indicated that these interventions can promote personal recovery [49].

Previous studies have proposed explanations for a close connection between illness self-management and personal recovery [5]. Long periods with serious symptoms, relapses, and stresses can affect the sense of well-being. Therefore, better symptom management is required to develop hope for the future and provide more opportunities to work on personal recovery goals [5]. Helpful in learning to cope with symptoms is teaching patients basic facts concerning their mental illness and working together on relapse prevention planning [5, 50]. Taking more control of their own lives promotes empowerment and the rebuilding of a positive sense of identity [5, 51]. Learning illness management also entails gaining social support, including peer support, and greater involvement with family members. This may contribute to a sense of connectedness [5, 51]. This also applies to working towards meaningful activities. All mentioned components of illness self-management are measured with the IMR scale. All mentioned components of personal recovery are included in the CHIME conceptual framework.

The conclusion that improvement in clinical and functional recovery may not be a prerequisite for improvement in personal recovery is supported by the following previous studies: First, our cross-sectional study–as mentioned in the introduction–showed that for predicting personal recovery, the ability to cope with symptoms (i.e. self-management) is more relevant than the symptoms themselves [11]. Second, in our earlier RCT we found a statistically significant effect of IMR + care as usual versus care as usual alone in self-reported overall illness management, as well as in personal recovery [12]. However, in this RCT, we found no statistically significant effects in clinical and functional recovery [12]. Third, two previous meta-analyses on people with psychotic disorders indicated a small to moderate association between clinical and personal recovery and showed that personal recovery was explained only partly by symptom severity [16, 25]. The authors of one of these meta-analyses indicated that their

findings imply that patients who still experience symptoms, might nevertheless report good personal recovery [16]. The same meta-analysis concluded therefore that treatment and outcome monitoring of patients with schizophrenia spectrum disorders should separately pay attention to clinical and personal recovery [16]. Fourth, a previous systematic review and meta-analysis (N = 845) showed the efficacy of the Wellness Recovery Action Plan (WRAP) for improving personal recovery without improving clinical recovery outcomes [27]. Fifth, several authors affirm the relevance of improvement in personal recovery and emphasize the importance of the therapeutic benefits of managing mental illness thereby contributing to personal well-being and self-perceived growth, as well as the value of leading a fulfilling life despite the presence of persistent symptoms [8, 9, 13, 19]. For instance, over and above symptoms [8], the loss of self-esteem is a recognized component of the major impact of schizophrenia on an individual's life and self-image [52–54]. The concept of self-esteem corresponds to the concept of identity (i.e., a positive sense of self), which is a relevant constituent of personal recovery [12, 21].

In the present analysis, clinical recovery as a mediator weakly affects the relationship between illness self-management and personal recovery. This is consistent with the findings of two recent meta-analyses on people with psychotic disorders. Both meta-analyses indicated a small to moderate association between clinical and personal recovery [16, 25].

That functional recovery could also be a mediator between illness management and personal recovery as shown in the current analysis, is a variant of the IMR framework. However, functional recovery as a mediator only weakly affects the relationship between illness self-management and personal recovery. This might have been affected by the weak association between functional and personal recovery shown in two meta-analyses. One of these two meta-analyses showed a small positive association between general functioning, as measured by the Global Assessment of Functioning scale, and personal recovery [16]. The other meta-analysis showed small positive associations between social support, work and housing, and psychosocial functioning with personal recovery [25].

To include the effect of change over time in the analysis, we chose difference scores over raw post-intervention scores.

The current analysis concerned improvements in illness self-management (IM) measured with the IMRS client version. All study participants (both the intervention and the control group) were included, with measurements at both the first and third time points. Therefore, we were able to measure the association of IM improvement with the improvement of other concepts. We consider the impact of illness self-management in this analysis as support for the relevance of IMR, as IMR works to improve illness self-management.

Given that in this study, improvements in self-management predicted improvements in personal recovery more directly than indirectly, we suggest that it supports the relevance of self-management interventions such as IMR. Therefore, in agreement with Lean et al., we suggest that these interventions should form part of the standard care for people with SMI and be given greater priority in guidelines [3].

## Strengths

This mediation analysis has a firm base [45] because the hypothesized directions of associations between the concepts examined were derived from theory [4], from previous research results by other researchers [3, 16, 25], and from ourselves [12]. This might support the relevance of the present outcomes [45]. Moreover, in this study, a possible different working of IMR has been suggested than that previously proposed [4]. This concerns introducing functional recovery as a possible mediator between illness management and personal recovery.

Two meta-analyses have respectively investigated the impact of clinical and functional recovery on personal recovery, but not the impact of illness management [16, 25]. A separate meta-analysis examined the impact of illness self-management [3]. However, in this study, both the indirect role of clinical and functional recovery and the direct role of illness management as determinants of personal recovery were investigated.

Our RCT on IMR, from which the data of this mediation analysis were derived, was conducted in a natural setting [12]. Therefore, we suggest that the results of the current analysis may be generalizable.

## Limitations

In this study, by using a mediation model, we explored empirical support for the association between changes in the different concepts in the direction described in the adapted conceptual framework on the working mechanisms of IMR. The original RCT from which the data derived included three time points of measurement. However, to be able to empirically test the adapted IMR conceptual framework using a longitudinal model, for instance by using a random intercept cross-lagged panel model [55], required measurements at four time points. This is because the proposed working of IMR begins with a suggested change in illness self-management (M1-M2), which could cause a change in functional and clinical recovery (M2-M3), which in turn could cause a change in personal recovery (M3-M4). Therefore, by only using the data of the three available measurement points, we were not able to execute a longitudinal analysis of the total adapted conceptual framework.

Another limitation is that all outcomes used were self-report questionnaires, which are inherently subjective. This may have caused bias in the associations between the measured concepts. Therefore, future research can also use clinician-rated outcomes, such as the clinician-rated IMR-Scale measuring illness self-management [33], the Brief Psychiatric Rating Scale for clinical recovery [56], and the Social and Occupational Functioning Assessment Scale for functional recovery [57].

Personal recovery should be self-reported. In the present study, the MHRM was used, which is a composite personal recovery scale. However, as there are various measures for personal recovery [25], future research could use another scale for measuring personal recovery, such as the Recovery Assessment Scale [58].

## Conclusions

This study's results showed empirical support for the association between illness self-management and personal recovery. Moreover, self-reported illness management appeared to be a stronger direct determinant of personal recovery than indirectly via clinical and functional recovery. Furthermore, this study confirmed that improvements in clinical and functional recovery may not be prerequisites for improvements in personal recovery. Therefore, this study supports the relevance of self-management interventions such as IMR. These mental health services deserve more attention and should form part of the standard mental health care for people with SMI.

The results partly provide empirical support for the working of IMR, as suggested in the conceptual framework. However, in this study, a possible new variant of this IMR framework was demonstrated, because functional recovery could be a mediator between illness management and personal recovery.

To further investigate our results, future research using a longitudinal analysis and other outcome-measures for measuring the same concepts is recommended.

## Supporting information

**S1 Checklist. AGReMA-SF checklist.**
(PDF)

**S2 Checklist. PLOS ONE clinical studies checklist.**
(DOCX)

**S3 Checklist. STROBE statement.** Checklist of items that should be included in reports of observational studies.
(DOCX)

**S1 Fig. The original conceptual framework of the IMR program (4).**
(JPG)

## Acknowledgments

We thank the patients, clinicians, and staff at Parnassia Groep and Yulius Mental Health, who participated in this study.

## Author Contributions

**Conceptualization:** Bert-Jan Roosenschoon, Mathijs L. Deen, Jaap van Weeghel, Astrid M. Kamperman, Cornelis L. Mulder.

**Data curation:** Bert-Jan Roosenschoon, Mathijs L. Deen.

**Formal analysis:** Bert-Jan Roosenschoon, Mathijs L. Deen.

**Funding acquisition:** Bert-Jan Roosenschoon, Cornelis L. Mulder.

**Methodology:** Bert-Jan Roosenschoon, Mathijs L. Deen, Jaap van Weeghel, Astrid M. Kamperman, Cornelis L. Mulder.

**Supervision:** Jaap van Weeghel, Astrid M. Kamperman, Cornelis L. Mulder.

**Writing – original draft:** Bert-Jan Roosenschoon, Mathijs L. Deen.

**Writing – review & editing:** Bert-Jan Roosenschoon, Mathijs L. Deen, Jaap van Weeghel, Astrid M. Kamperman, Cornelis L. Mulder.

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
