## [Decision Letter · Decision Letter 0]

6 Sep 2023

PONE-D-23-14969Illness (self) management, clinical and functional recovery as determinants of personal recovery in people with severe mental illnesses: A mediation analysisPLOS ONE

Dear Dr. Roosenschoon,

Thank you for submitting your manuscript to PLOS ONE. After careful consideration, we feel that it has merit but does not fully meet PLOS ONE’s publication criteria as it currently stands. Therefore, we invite you to submit a revised version of the manuscript that addresses the points raised during the review process.

The manuscript has been evaluated by three reviewers, and their comments are available below.

The reviewers have raised a number of concerns that need attention. They request additional information on methodological aspects of the study (such as the inclusion of information on the sample size and response rate), revisions to the statistical analyses and they question the internal and external validity of the results reported.

Could you please revise the manuscript to carefully address the concerns raised?

We look forward to receiving your revised manuscript.

Kind regards,

Avanti Dey, PhD

Senior Staff Editor

PLOS ONE

Reviewers' comments:

Reviewer's Responses to Questions

**Comments to the Author**

1. Is the manuscript technically sound, and do the data support the conclusions?

Reviewer #1: Partly

Reviewer #2: Yes

Reviewer #3: Yes

2. Has the statistical analysis been performed appropriately and rigorously? 

Reviewer #1: No

Reviewer #2: Yes

Reviewer #3: Yes

3. Have the authors made all data underlying the findings in their manuscript fully available?

Reviewer #1: No

Reviewer #2: Yes

Reviewer #3: Yes

4. Is the manuscript presented in an intelligible fashion and written in standard English?

Reviewer #1: Yes

Reviewer #2: Yes

Reviewer #3: Yes

5. Review Comments to the Author

Reviewer #1: Thank you for the opportunity to review this manuscript on the relationship between change in illness management skills and different facets of recovery among people with serious mental illnesses. This is a well written manuscript with a number of methodological strengths. The RCT from which the data are derived is well described and appears to be methodologically sound. Further, the authors use established and widely used measures of illness self-management, and personal, clinical, and functional recovery that have been validated in samples of people with serious mental illnesses. Finally, the sample size appears sufficient to support the analyses.

There are two major issues that limit enthusiasm for the manuscript in its current form. First, it is not clear why the researchers did not assess change in the constructs of interest in longitudinal models given that there were three data collection time points. This would be a vast improvement over using change scores given problems with reliability and interpretability of change across only two time points. Second, the fact that the sample included participants assigned to both the Illness Management and Recovery and usual care conditions makes results interpretation difficult. The general conclusion seems to be that Illness Management and Recovery is indicated to promote personal recovery, but the analyses were not set up to support this claim. The researchers might consider limiting their sample to only those who participated in IMR, or, at the very least, entering treatment condition as a covariate into the mediation models.

Minor issues:

Please state what conceptual framework is being used to support the study. CHIME is mentioned, but it is not clear that this is the framework on which the study is based.

The authors might consider citing Thomas et al.’s (2018) meta-analysis of person-oriented recovery interventions, which found that interventions including IMR were associated with improvement in personal recovery.

Were participants screened for eligibility based on self-report?

To enhance interpretability of study findings, descriptive statistics pertaining to the study measures at each time point should be provided.

Suggest reframing the results section to more directly answer the stated research question: 1) the association of changes over time in the illness self-management skills of people with SMI on changes in personal recovery, and 2) the degree to which this association is mediated by changes in clinical and functional recovery.

I am not sure that functional recovery as a mediator between illness management and personal recovery reflects a variant of the IMR framework, as functional recovery is sometimes considered as synonymous with clinical recovery (see: Liberman et al. 2002).

Reviewer #2: Dear Author(s),

I commend your work on the association between illness self-management skills and personal recovery in SMI patients. Your methodological rigor, especially the bootstrap techniques and mediation analysis, is admirable.

Abstract: Consider moving the "clinical trial registration number" to the methods section if it doesn't add critical value in the abstract.

Introduction:

-Elaborate briefly on previous study methodologies and findings.

-Clarify the shift from prior cross-sectional studies to the current RCT.

-Expand on theoretical or empirical evidence that suggests potential mediating relationships.

Method: Address the discrepancy in the 3:2 ratio of IMR + CAU to CAU alone participants. Ensure clarity on randomization procedures and reasons for any deviations.

Measures: Provide more details on the reliability, validity, and potential translation (English to Dutch) of the tools used. Elaborate on the IMR program's components, objectives, and methodologies, considering its centrality to the study.

Results:Discuss the choice and implications of using difference scores versus raw post-intervention scores.

Discussion:

-Provide more supporting evidence for strong claims.

-Explore why clinical and functional recovery had 'weak' mediating effects.

-Discuss similarities or differences with prior studies more explicitly.

-Elaborate on how the results specifically support the relevance of IMR.

-Address potential contradictions, e.g., the role of symptom management in developing hope for the future.

Your manuscript provides valuable insights. With the proposed enhancements, it can offer even greater depth and clarity to readers. Ther's no any concerns about dual publication, research ethics, or publication ethics.

Warm regards

Reviewer #3: The authors present results of a mediation analysis of data from a clinical trial in individuals with severe mental illness. In particular, they assessed whether changes in clinical recovery and functional recovery were mediators of the association between change in illness self-management (IM) and change in personal recovery. There mediation model is informed by theory, prior results and meta-analyses, so has justification. The authors identified both direct and indirect effects of IM on personal recovery, though the direct effect was much stronger. The manuscript is well written. I have some minor comments that would provide further clarification.

1. In lines 30 and 201, authors state that difference scores are from subtracting means at baseline from means at the final time point for the change measures. Aren't these computed at the individual level? It is not clear what the "means" would be for each person. Certainly, at the group level, the average difference score would be the difference in means, but not at the individual level. Authors should clarify what is done.

2. In lines 175-176, authors indicate that IM has 3 subscales. It appears as though authors just use a single measure (at 2 time points) for IM - is this a total score? If so, authors should state that even though the scale has subscales, a total score was used for analysis (or if subscales were used, that should be clarified).

3. In line 216, authors refer to a regression equation from delta IM to delta PR as the direct effect. That is the path, but the regression equation actually includes the mediators, right?

6. PLOS authors have the option to publish the peer review history of their article (what does this mean?). If published, this will include your full peer review and any attached files.

Reviewer #1: No

Reviewer #2: **Yes: **Jutharat Thongsalab

Reviewer #3: No

---

## [Author Response · Author response to Decision Letter 0]

12 Jul 2024

Dear reviewer #2. Sometimes you asked for brief, but comprehensive revisions. However, it required a lot of space to mention all sample size, methodologies and results of all cited literature! Therfore sometimes I had to choose for brief!

---

## [Decision Letter · Decision Letter 1]

22 Oct 2024

Illness (self) management, clinical and functional recovery as determinants of personal recovery in people with severe mental illnesses: A mediation analysis

PONE-D-23-14969R1

Dear Dr. Roosenschoon,

We’re pleased to inform you that your manuscript has been judged scientifically suitable for publication and will be formally accepted for publication once it meets all outstanding technical requirements.

Kind regards,

Alessandro Rodolico

Academic Editor

PLOS ONE

Additional Editor Comments (optional):

The authors have provided adequate comments and responses to the reviewers' requirements. The manuscript has now been improved sufficiently for publication. Regarding my final comments, I would suggest only moving the study objectives in the abstract to be placed next to the rationale.

Reviewers' comments:

Reviewer's Responses to Questions

**Comments to the Author**

1. If the authors have adequately addressed your comments raised in a previous round of review and you feel that this manuscript is now acceptable for publication, you may indicate that here to bypass the “Comments to the Author” section, enter your conflict of interest statement in the “Confidential to Editor” section, and submit your "Accept" recommendation.

Reviewer #1: All comments have been addressed

Reviewer #2: All comments have been addressed

2. Is the manuscript technically sound, and do the data support the conclusions?

Reviewer #1: (No Response)

Reviewer #2: Yes

3. Has the statistical analysis been performed appropriately and rigorously? 

Reviewer #1: (No Response)

Reviewer #2: N/A

4. Have the authors made all data underlying the findings in their manuscript fully available?

Reviewer #1: (No Response)

Reviewer #2: Yes

5. Is the manuscript presented in an intelligible fashion and written in standard English?

Reviewer #1: (No Response)

Reviewer #2: Yes

6. Review Comments to the Author

Reviewer #1: (No Response)

Reviewer #2: 1. I suggested moving the clinical trial registration number to the methods section, and this recommendation was followed in the revised manuscript.

2. I asked for more elaboration on the transition from cross-sectional studies to the current RCT and details on the mediating relationships. These suggestions were addressed, as the revised version provides more clarity.

3. The issue of a 3:2 randomization ratio was raised. The revised version briefly addresses this but does not go into great depth about the reasons behind the deviation from randomization.

4. I asked for more details about the validity and reliability of the tools used. This was addressed more thoroughly in the revised version, including additional explanation of the measures.

5. The revised manuscript still uses different scores despite reviewer concerns. The authors explained their rationale but did not change this methodological choice.

6. I suggested strengthening the discussion of the weak mediating effects of clinical and functional recovery. This was partially addressed in the revised manuscript.

7. PLOS authors have the option to publish the peer review history of their article (what does this mean?). If published, this will include your full peer review and any attached files.

Reviewer #1: No

Reviewer #2: No

---

## [Editor Report · Acceptance letter]

14 Nov 2024

PONE-D-23-14969R1 

PLOS ONE

Dear Dr. Roosenschoon, 

I'm pleased to inform you that your manuscript has been deemed suitable for publication in PLOS ONE. Congratulations! Your manuscript is now being handed over to our production team.

Kind regards, 

on behalf of

Dr. Alessandro Rodolico 

Academic Editor

PLOS ONE